# Treeline displacement may affect lake dissolved organic matter processing at high latitudes and altitudes

Núria Catalán [1,7] ✉, Carina Rofner[2], Charles Verpoorter[3], María Teresa Pérez[2], Thorsten Dittmar[4,5], Lars Tranvik [1], Ruben Sommaruga [2] & Hannes Peter [2,6] ✉

Climate change induced shifts in treeline position, both towards higher altitudes and latitudes induce changes in soil organic matter. Eventually, soil organic matter is transported to alpine and subarctic lakes with yet unknown consequences for dissolved organic matter (DOM) diversity and processing. Here, we experimentally investigate the consequences of treeline shifts by amending subarctic and temperate alpine lake water with soil-derived DOM from above and below the treeline. We use ultra-high resolution mass spectrometry (FT-ICR MS) to track molecular DOM diversity (i.e., chemodiversity), estimate DOM decay and measure bacterial growth efficiency. In both lakes, soil-derived DOM from below the treeline increases lake DOM chemodiversity mainly through the enrichment with polyphenolic and highly unsaturated compounds. These compositional changes are associated with reductions in bulk and compound-level DOM reactivity and reduced bacterial growth efficiency. Our results suggest that treeline advancement has the potential to enrich a large number of lake ecosystems with less biodegradable DOM, affecting bacterial community function and potentially altering the biogeochemical cycling of carbon in lakes at high latitudes and altitudes.

Treeline ecotones, the transition zones between dense forests and low-stature alpine or arctic vegetation, reflect the ecological forces exerted by a combination of regional climate, topography, and disturbance history[1]. The treeline roughly follows mean growing season air temperature isoclines, however, soil temperature, and particularly near-surface permafrost better explain local treeline position[2]. The position of the treeline is not static, it is sensitive to land use practices and climatic changes, such as variations in temperature, precipitation, and snowpack duration[3–6]. Consequently, global change is currently shifting and expanding the treeline towards higher altitudes and latitudes[7], with temperature increase being the main climatic driver in mountain treelines of boreal regions[8].

The transition from tundra to boreal forest and the upward shift of the coniferous treeline in temperate mountain regions is accompanied by shifts in inorganic soil nutrients and soil organic matter content and composition[9–11]. Spurred by permafrost thaw, earlier snow melt and an expected increase in precipitation due to climate change[12], this will affect the carbon export from catchments to aquatic

[1]Limnology, Department of Ecology and Genetics, University of Uppsala, Uppsala, Sweden. [2]Lake and Glacier Ecology Research Group, Department of Ecology, Universität Innsbruck, Innsbruck, Austria. [3]University Littoral Côte d'Opale, Centre National de la Recherche Scientifique (CNRS), Université Lille, IRD, UMR -LOG-Laboratoire d'Océanologie et de Géosciences, F-Wimereux, France. [4]Institute for Chemistry and Biology of the Marine Environment (ICBM), Carl von Ossietzky Universität Oldenburg, Oldenburg, Germany. [5]Helmholtz Institute for Functional Marine Biodiversity (HIFMB) at the Carl von Ossietzky Universität Oldenburg, Oldenburg, Germany. [6]River Ecosystems Laboratory, École Polytechnique Fédérale de Lausanne (EPFL), Lausanne, Switzerland. [7]Present address: SHE2, Institute of Environmental Assessment and Water Research (IDAEA), CSIC, Barcelona, Spain. ✉e-mail: nuria.catalan@idaea.csic.es; hannes.peter@epfl.ch

ecosystems in mountain areas of high altitudes and latitudes[13], with potential consequences for the biogeochemical processing of dissolved organic matter (DOM) in lakes. Indeed, paleolimnological evidence suggests that lake productivity[14] and biogeochemistry[15] are sensitive to changes in terrestrial DOM subsidies linked to treeline position. Despite the global scale of this phenomenon, the knowledge on the consequences of treeline shifts for DOM composition and processing in alpine and subarctic lakes is very scarce. This knowledge, however, is crucial, particularly because of the large number of lakes potentially affected. Moreover, lakes at high altitudes and latitudes are particularly sensitive to global change[16,17] as well as key drinking water resources[18] and modifications in the sources and nature of terrestrial carbon inputs, particularly those that affect DOM reactivity, may alter the role of these lakes in the global carbon cycle.

The taiga-tundra transition zone stretches for more than 13,400 km around the northern hemisphere and current poleward treeline advance rates range from 10 to 100 m per year[4]. Most lakes larger than 0.01 km² are located at latitudes above 60°N, and 15% of all lakes globally are located at elevations above 500 m above sea level[19], suggesting that a large number of lakes may be affected by the advance of the treeline. Together with the vulnerability of lakes to climate change[20–23], alterations in their allochthonous DOM loading linked to treeline displacement may impact their biogeochemical function by channeling terrestrial C into the atmosphere, altering regional and global C budgets[24,25]. The magnitude of this lateral C flux will be affected by DOM composition and reactivity.

To assess the impact of treeline displacement on DOM composition and reactivity in high-altitude and latitude mountain lakes, we combined in situ and laboratory experiments. We selected a subarctic catchment in northern Finland (679 m a.s.l., 69.05°N) and an alpine one in Austria (2417 m a.s.l., 47.23°N), both located in discontinuous/sporadic permafrost areas. To simulate the effects of treeline shifts on DOM biogeochemistry, we amended lake water with superficial soil-extracted DOM from above (S-Above) and below (S-Below) the treeline from the same region and incubated it in situ in mesocosms for 72 h. In parallel, we conducted long-term (81 d) laboratory incubations to assess DOM degradation kinetics and bacterial growth efficiency. We tracked DOM compositional change on a molecular formula level using Fourier-transform ion cyclotron mass spectrometry (FT-ICR MS; "Methods") and derived estimates of DOM chemodiversity and reactivity. We hypothesized that DOM from below the treeline will enrich lake water chemodiversity. However, there may not be an increased reactivity because the below-treeline DOM may be inherently less available to microbial degradation compared with DOM derived from above the treeline. Taken together, we expected that the experimental manipulations would result in reduced DOM processing in lakes.

## Results and discussion

### Lake DOM composition altered as a function of soil origin

Soil extract additions altered DOM composition at the compound level as a function of soil origin. However, between 85 and 88% of the >2500 DOM molecular formulae detected by FT-ICR MS were initially present in the water of both lakes (Fig. 1, Table 1). Based on comparison with previous studies on DOM chemodiversity[26,27], we suggest that these compounds can be considered ubiquitous in lake water and are probably the result of microbial transformation processes of DOM of various origins[27]. The remaining 12–15% of DOM compounds were only found in amended samples and were thus considered as derived directly from soil. Compared to the original lake water, 23–28% of the molecular formulae were novel or enriched (in terms of FT-ICR MS signal intensity) in treatments amended with soil extracts from above the treeline, and 44–50% of the compounds were novel or enriched when soil extract from below the treeline was added to the lake water. This was similar for both the alpine lake (28% in S-Above and 44% in

S-Below treatments, respectively) and the subarctic lake (23% in S-Above and 50% in S-Below treatments, respectively).

In both regions, soil extract from below the treeline increased the relative contribution of polyphenolic and highly unsaturated DOM compounds, while in the S-Above treatment, the proportion of unsaturated and saturated compounds increased (Fig. 1, Table 1). Those compound categories are defined based on the H:C and aromaticity of the compounds and are used to highlight key chemical differences and facilitate molecular formulae interpretation (see "Methods"). Differences between S-Above and S-Below were particularly pronounced in the subarctic lake (Figs. 1c and d), where the relative contribution of highly unsaturated compounds in S-Below was two times higher than that in S-Above (Table 1). In that lake, the relative contribution of unsaturated compounds in S-Above was three times higher than in S-Below. Furthermore, the enrichment of heteroatomic features (N, P, and S containing compounds), related to fresh DOM and microbial processes in lakes[28], was higher in the S-Above than in the S-Below samples for both lakes (Table 1). These findings were supported by fluorescence spectroscopy measurements, which, for the S-Below treatment, showed a marked increase in the relative contribution of the humic-like fluorescence components, previously related to terrestrial sources[29], see Supplementary Figs. S1 and S2. In both lakes, the S-Above treatment increased the proportion of protein-like fluorescence related to in situ DOM sources (ANOVA, $F_{Treatment} = 21.91$, $p$-value < 0.001).

### Soils below treeline add less chemodiversity to lake water

To address how added DOM formulae integrate into the already existing lake chemodiversity, we developed a measure of chemodiversity which incorporates chemical similarity ("Methods"). Given the numerous interactions between microbes and the organic matter pool, DOM chemodiversity is expected to influence ecosystem functioning and biogeochemical cycles[30,31]. Moreover, our approach leverages random sampling of a fraction of compounds, allowing for an unbiased comparison of chemodiversity beyond the number of molecules (Fig. 2). Hence, we accounted for chemical similarity of DOM moieties, whereby compounds with similar H:C and O:C ratios, molecular weight, double bond equivalents, modified aromaticity index and heteroatoms, contribute relatively less to chemical diversity than compounds with divergent values of those compounds' characteristics (i.e., chemically novel compounds). With this considered, the addition of soil extract reduced chemodiversity in the S-Above treatment for both regions (on average by 22.7% in the alpine and by 9.8% in the subarctic region, Fig. 2). This observed reduction in DOM chemodiversity indicates that local soil extracts preferentially contribute chemically similar moieties to those already present in the lake water DOM pool. In the S-Below treatment, soil extracts reduced chemodiversity by 10.0% on average in the alpine region but increased chemodiversity by 2.9% in the subarctic region (Fig. 2). During the in situ incubations, chemodiversity in the alpine lake increased continuously in S-Above treatments (on average by 31.6%, Fig. 2), while the S-Below treatment did not result in consistent temporal changes in chemodiversity. In contrast, chemodiversity did not change over time in any treatment of the subarctic lake (Fig. 2b). Collectively, this suggests that soil-derived organic matter provided novel molecular entities to the DOM pool in both the subarctic and alpine lake located above treeline. Whereas soil-derived DOM from local soils (i.e., above treeline) appears chemically nested within lake water DOM, soil extracts from below the treeline introduce chemically novel moieties to the lakes.

### Less reactive DOM from soils below the treeline

To assess the overall reactivity of DOM, we studied its degradation kinetics in long-term degradation experiments (see "Methods" and Supplementary Fig. S3). In both lakes, reactivity continuum models fitted to DOC decay curves showed that samples amended with DOM

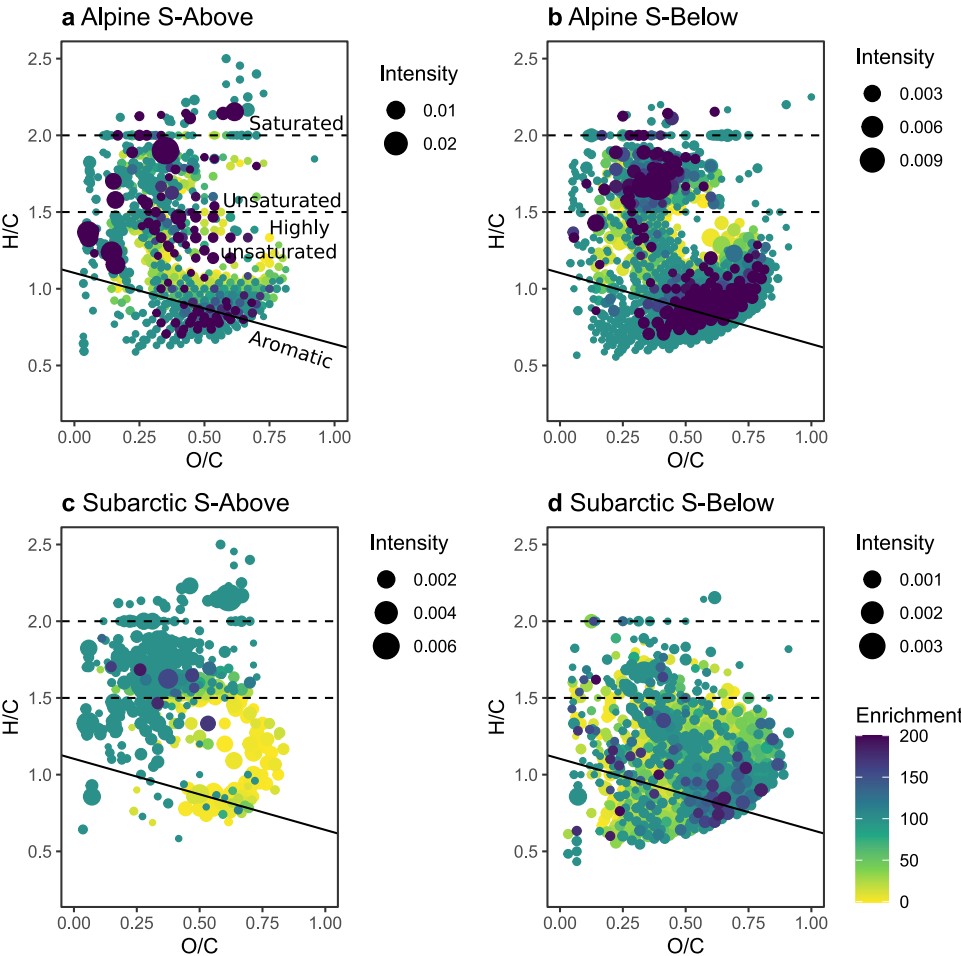

**Fig. 1 | Enriched compounds in the different soil exudate amendments.** Panels **a**–**d** show Van Krevelen diagrams of compounds enriched upon soil exudate amendment from above (S-Above) and below (S-Below) the treeline for the sub-arctic and alpine lakes. Colors represent enrichment, with, compared to control treatments, newly detected compounds being enriched at 100% and a common scale across panels. Enrichment larger than 100% indicates compound increase compared to control, while enrichment smaller than 100% indicates compound decrease compared to control. Symbol size corresponds to mean intensity. Lines separate compound categories, indicated in panel (**a**). Source data can be found on zenodo under https://doi.org/10.5281/zenodo.10578476.

## Table 1 | Molecular characteristics of the two lakes and of the enriched compounds for each treatment

| | Alpine | | | Subarctic | | |
|---|---|---|---|---|---|---|
| | Lake water | S-Above *vs* Lake | S-Below *vs* Lake | Lake water | S-Above *vs* Lake | S-Below *vs* Lake |
| Total number of compounds | 2631 | | | 2556 | | |
| Total number of enriched compounds[a] | | 904 | 1445 | | 631 | 1389 |
| Total "new" compounds[b] | | 397 | 767 | | 335 | 383 |
| N-containing compounds[a] | 944 | 120 (13) | 128 (9) | 678 | 76 (12) | 132 (10) |
| P-containing compounds[a] | 284 | 26 (3) | 60 (4) | 137 | 66 (10) | 14 (1) |
| S-containing compounds[a] | 372 | 167 (18) | 109 (8) | 339 | 132 (21) | 134 (10) |
| Aromatics[a] | 286 | 154 | 258 | 516 | 36 | 322 |
| Highly unsaturated[a] | 1645 | 466 | 753 | 1649 | 155 | 932 |
| Unsaturated[a] | 580 | 195 | 349 | 369 | 387 | 119 |
| Saturated[a] | 102 | 89 | 85 | 22 | 53 | 16 |
| Average DBE[a] | 7.87 | 8.47 | 11.06 | 9.28 | 5.31 | 11.67 |
| Average O/C[a] | 0.45 | 0.41 | 0.43 | 0.43 | 0.39 | 0.49 |
| Average H/C[a] | 1.32 | 1.30 | 1.20 | 1.20 | 1.57 | 1.09 |
| Average AI$_{mod}$[a] | 0.24 | 0.28 | 0.31 | 0.31 | 0.13 | 0.34 |

[a]For those variables, values correspond to all the compounds in the case of lake waters and only to the enriched compounds in the treatments.
[b]Total "new" compounds represent only those compounds that were present in the treatment but not in the corresponding lake water. Compound classes, according to Merder et al.[55] were "saturated" (H/C ≥ 2), "unsaturated" (H/C ≥ 1.5), "highly unsaturated" (H/C < 1.5, AI$_{mod}$ < 0.5) and "aromatics" (0.5 < AI$_{mod}$ < 0.67).

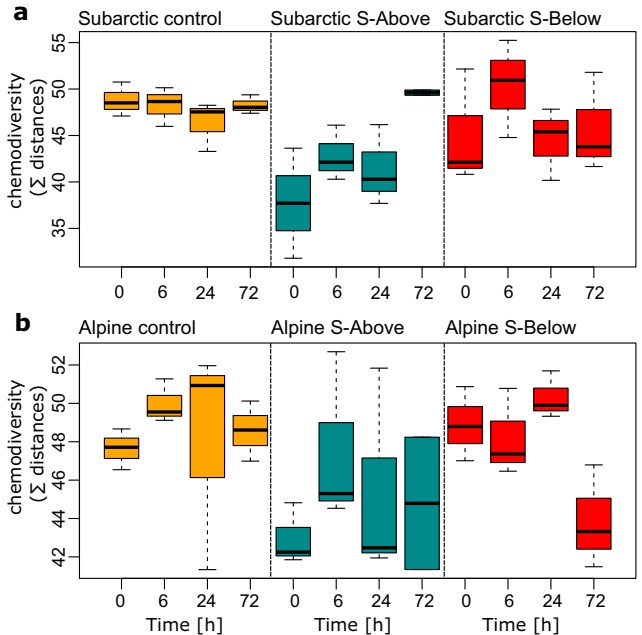

**Fig. 2 | DOM chemodiversity assessed as the pathlength in a dendrogram of chemical similarity of molecular formulae detected via FT-ICR MS.** Boxplots showing median (horizontal lines), interquartile ranges (IQR, boxes) and whiskers extending to 1.5 × IQR of chemodiversity estimates ($n = 71$) for each timepoint for the subarctic (**a**) and alpine (**b**) lake, respectively. Note the initial drop in chemodiversity in soil-amended treatments, particularly in S-Above treatments, indicating the nested diversity that local soil-derived organic carbon contributes to lake water DOM. While chemodiversity was temporally relatively stable, particularly S-Above treatments converged to more similar levels of chemodiversity as compared to the controls, reflecting the effects of microbial degradation of amended organic matter. Source data can be found on zenodo under https://doi.org/10.5281/zenodo.10578476.

from above the treeline exhibited a higher reactivity, with lower apparent age of the most reactive compounds and lower relative predominance of the most recalcitrant compounds (i.e., α and v parameters Supplementary Fig. S3). Those parameters were in line with the initial decay rates, higher in the amendment of soil above the treeline for the subarctic lake although not in the alpine lake (Supplementary Table S1).

To assess how the carbon metabolism of bacteria may be affected by treeline shift associated changes in soil-derived DOM, we measured bacterial carbon production and respiration and estimated bacterial growth efficiency (BGE) in the in situ experiments. Bacterial production strongly depended on location (linear mixed effects model ANOVA (lme ANOVA), F = 41.8, $p$-value < 0.001), but in both cases followed similar patterns, with S-Above sustaining higher bacterial production than S-Below (Fig. 3b; lme ANOVA, F = 39.87, $p$-value < 0.001). The interaction between location and treatment was not significant, indicating that the effect of the soil amendments was similar in both the alpine and subarctic location (lme ANOVA, F = 0.95, $p$-value > 0.05). Soil extract amendment also had significant positive effects on bacterial respiration (lme ANOVA, F = 5.09, $p$-value = 0.007), but the strength of those effects depended on location (lme ANOVA, F = 13.31, $p$-value < 0.001). This was reflected in bacterial growth efficiency, which was significantly higher in the subarctic than in the alpine site (lme ANOVA, F = 58.11, $p$-value < 0.001). Thus, lake bacterial communities produced more biomass per unit of C assimilated in the S-Above than in the S-Below treatments (lme ANOVA, F = 24.94, $p$-value < 0.001). However, the magnitude of this effect differed between both locations (lme ANOVA, F = 3.52, $p$-value = 0.03). Overall, these results suggest that by shifting allochthonous DOM sources towards lower

reactivity, treeline advancement has the potential to affect ecosystem functioning by reducing bacterial growth efficiency in both alpine and subarctic lakes.

Finally, by using changes in relative signal intensity of individual DOM molecular formulae in reactivity continuum models, we obtained compound-specific decay and increase rates (Fig. 3 and Supplementary Figs. S6,7). The distribution of compound-specific degradation and production rates was skewed towards negative rates (i.e., degradation) in both controls and treatments (Supplementary Fig. S6 and Supplementary Table S2), except for the alpine S-Below amendment. However, unamended controls and S-Above treatments contained compounds with the highest estimated decay rates, particularly saturated, unsaturated, and oxygen-poor highly unsaturated moieties (Fig. 3a). In contrast, only a few aromatic and polyphenolic compounds followed a first-order decay model (i.e., it was impossible to fit them, either because they follow other kinetics or because they did not change). In agreement with this, the compounds that positively correlated with decay rates (i.e., faster decay) obtained from the reactivity continuum models included unsaturated aliphatic and saturated fatty acids, as well as oxygen-rich phenolic and highly unsaturated compounds (Fig. 3b). There are several examples of saturated compounds being more labile[32–34] and, although various DOM degradation pathways can be confounded, the apparent lability reported here can be attributed to biodegradation. These results thus indicate that soils above the treeline contribute DOM compounds that sustain high DOM decay rates. Those soils represent the DOM inputs that the lake is already receiving during the melting season or rain events. We cannot attribute compounds in the control treatments to allochthonous (i.e., soil-derived) or autochthonous (i.e., in situ production) origins, but the fact that the DOM addition from soils above the treeline enriched lake water with these compounds highlights their relevance as readily available carbon sources[35–37].

The addition of soil-derived DOM from below the treeline prompted the degradation of highly unsaturated phenolic compounds, especially apparent in the subarctic site (Fig. 3). However, also in the alpine site, the number of polyphenolic and aromatic compounds following first-order decay models, although low (see above), was substantially higher in S-Below than in S-Above treatments. Regarding compound enrichment (Fig. 1), we observed that in the alpine lake, S-Below addition increased both the proportion of unsaturated aliphatic and of polyphenolic compounds. In contrast, in the subarctic location, mainly polyphenolic compounds were enriched, corresponding well with the observed patterns of degradation (Fig. 3). The presence and reactivity of such compounds could be related to the elevated decay rates in the S-Below treatment attained at the end of the incubation (Supplementary Fig. S3). This agrees with the changes in the DOM bulk composition detected by optical measurements. Fluorescence-derived PARAFAC protein-like components C5 and C6 were abundant in the S-Above treatments in both locations. These components correspond to amino-acid-like and tyrosine-like organic matter and are related to unsaturated and highly unsaturated compounds, mainly oxygen-rich in the case of C6 (Supplementary Fig. S2). Therefore, shifts in the treeline have the potential to introduce chemically novel moieties that may be recalcitrant to bacterial degradation.

Our study concludes that shifts in the treeline and changes in soil organic matter inputs possess significant potential to influence the biogeochemistry of organic matter across numerous alpine and subarctic lakes. Specifically, our experimental findings offer initial evidence indicating that the advancing treeline may already be altering the chemical composition of dissolved organic matter (DOM) in lake water. However, it is evident that a comprehensive understanding of this intricate environmental change demands further work. Ideal next steps involve combining experimental efforts with in situ monitoring and remote sensing[9]. In addition, exploring other influential factors,

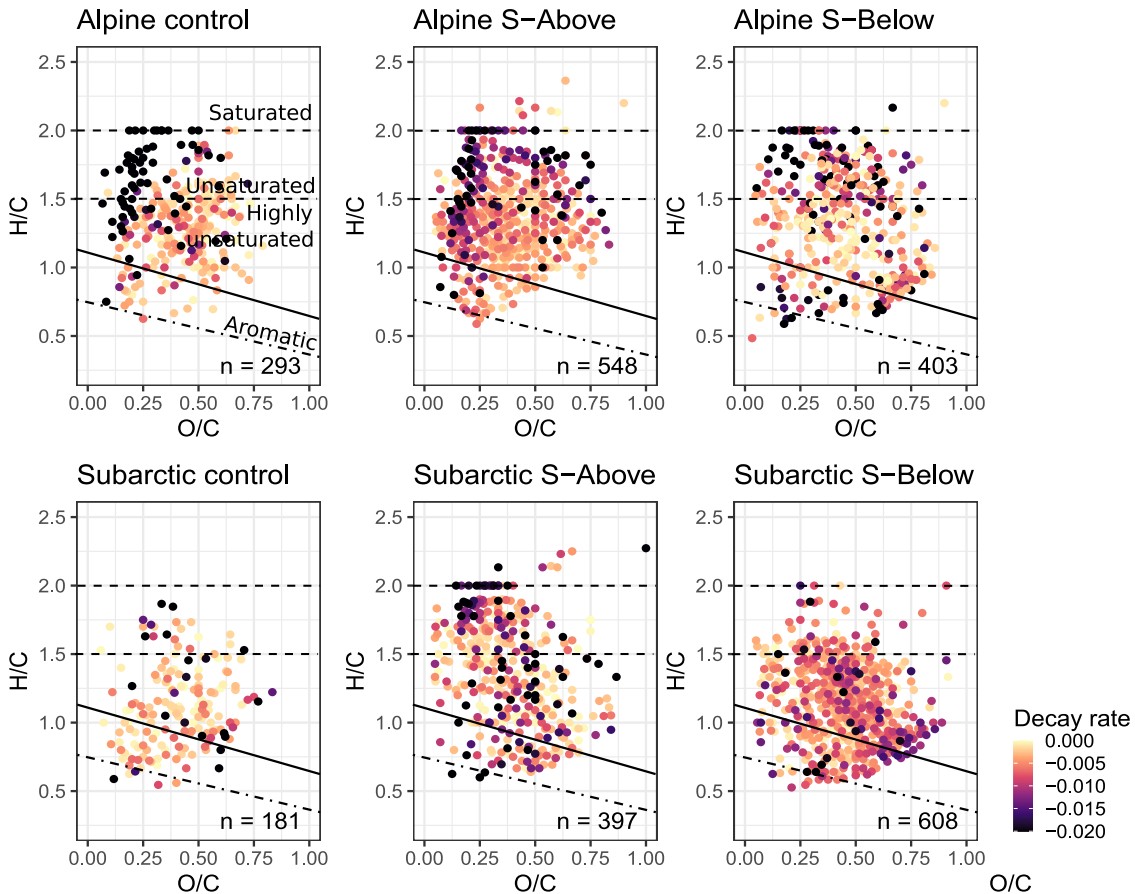

**Fig. 3 | Individual compound decay rates during in situ incubations and correlations with decay rates.** The Van Krevelen diagrams represent compounds showing a significant fit to first-order decay models for the control, S-Above and S-Below treatments for the subarctic and alpine lakes. Colors represent decay rates, with more negative values representing stronger degradation. Values shown are mean values across replicates, a compound was included only if the decay model was validated across all replicates. See Supplementary Figure S6 for distributions of decay rates across all the compounds and treatments. Source data can be found on zenodo under https://doi.org/10.5281/zenodo.10578476.

such as repeated freeze-thawing and DOM processing under winter conditions[38-40], photodegradation, and studies covering broader spatial and temporal scales, will be crucial.

Our research suggests that treeline shifts could introduce chemically novel DOM moieties and augment polyphenols and highly unsaturated compounds already present in lake water. These changes could extend beyond carbon biogeochemistry, potentially impacting drinking water quality and raising questions about the effects of such chemical novelty on resident microbial composition and function, subsequently influencing ecosystem health[41]. The consequences of these shifts might include negative impacts on DOM biodegradation rates and the efficiency of the bacterial communities in producing biomass, potentially disrupting alpine and subarctic lake food webs and carbon cycling. Yet, an intriguing possibility arises from the potential adaptation and evolution of microbial communities, both in catchment soils and lake water, at faster timescales than global changes[42]. Such adaptations could alter ecosystem-scale biogeochemical trajectories, offering a glimpse into potential resilience within these ecosystems amidst shifting environmental dynamics.

## Methods
### Study sites
We chose two model lake ecosystems for our experimental manipulations. Both lakes are well-studied long-term ecological monitoring sites. The subarctic, oligotrophic lake Saanajärvi is located at 679 m a.s.l. in north-western Finland (69°50N 20°870E). This medium-sized lake (70 ha) has a steep shore, a maximum depth of 24 m, and is ice-

covered for up to 9 months per year. The catchment (461 ha) mainly comprises steep mountain slopes and consists of bare rocks (limestone and calcareous dolomite) and a thin soil layer. Plant diversity is distinct from boreal forests and characterized by sparse mountain birch (*Betula pubescens*) forests, dwarf birch (*Betula nana*), lichens and subarctic flowering plants. The alpine lake Gossenköllesee is located at 2417 m a.s.l. in the Austrian Alps (47°130N 11°010E). Gossenköllesee is a small (1.7 ha), moraine-dammed seepage lake with a maximum depth of 9.9 m and usually ice-covered for up to 6–7 months per year. The catchment (30 ha) comprises crystalline bedrock, scarce alpine vegetation dominated by *Carex curvula*, various forb species and arbuscular lichens, and a poor soil layer. While both lakes are representative model systems for alpine and subarctic lakes in general, important differences, including, for example, catchment lithology, lake morphometry, and food-web structure contribute to the observed differences in carbon biogeochemistry. Moreover, the location of the treeline as well as treeline advancement, depends on a series of interconnected factors that change from global to local scales. At local scales, slope and aspect can significantly influence the treeline[43], which differ between the alpine and subarctic lake catchment. We therefore address high-level similarities in the responses of both lakes to soil-derived organic matter amendments.

### Experimental approach
During summer 2014, two short-term (3 days) in situ and two long-term (81 days) laboratory experiments were conducted in both the alpine and subarctic site. These durations were chosen based on

previous estimates of lake water DOM degradation (see ref. 44 for a review of DOC decay rates across systems).

In all experiments, lake water was amended with soil extracts from the same geographical region, namely, soil from the lake catchment (S-Above) and soil from a nearby catchment of a lake located below the treeline (S-Below). In the alpine sites, the S-Below soil was obtained from a subalpine forest dominated by *Picea alba* and *Pinus cembra* in the catchment of Piburgersee (47°11′N 10°53′E). In the subarctic region, the S-Below soil was derived from a spruce-dominated boreal forest (*Picea abies*) in the catchment of Bajit Ivgujärvi (69°21′N 20°19′ E). Bacteria-free soil extracts were prepared from the uppermost layer of soils according to Kalbitz et al.[45] (for details see ref. 46), resulting in extract concentrations of 80 mg C/L to 141.6 mg C/L. In the case of the short-term in situ experiments, water was incubated in dark mesocosms (10 L) submersed in the lakes for three days. For this, lake water from 1 m depth was filtered through 1.0 μm (Polycap 75 filter column, Whatman, GE Healthcare). Further details on the setup of in situ incubations are provided in ref. 46. Depending on DOC concentration in the lake (0.4 ± 0.09 mg C/L in Gossenköllesee and 1.5 ± 0.1 mg C/L in Saanajärvi) and the soil extracts, between 213 and 438 mL of soil extracts were added to 10 L of filtered lake water to increase the DOC concentration by a factor of 3. Triplicates were prepared for both soil extract additions and the control (no addition). At each timepoint during the short-term incubations (i.e., after 0, 6, 24, and 72 h), samples were collected for DOM characterization (FT-ICR MS and optical spectroscopy), bacterial carbon production (³H-leucine incorporation) and the quantification of DOC concentration.

In the long-term laboratory experiments, incubations were performed in 42 mL pre-combusted (450 °C for 4 h) glass vials in the dark at 20 °C for 81 days. For this, lake water was sequentially filtered through pre-combusted 1.2 μm filters (Whatman GF/C) and pre-rinsed 0.2 μm filters (nylon membrane filters, Whatman). Then, the corresponding soil extracts were added. Triplicate samples were incubated with an inoculum (1.2 μm filtered water of each corresponding lake) added in a 1:10 v/v proportion. This inoculum might underestimate the significance of fungi for degradation. Triplicates were incubated in a water bath in the dark at 20 °C. Individual vials were removed at each time step (i.e., after 0 h, 12 h, 1 d, 3 d, 5 d, 7 d, 12 d, 22 d, 40 d, 58 d, and 81 d). DOM was characterized by optical spectroscopy and DOC concentration measurement, bacterial activity and respiration rates were measured.

Our experimental simulations clearly address only a fraction of the potentially multifactorial effects of treeline advances on allochthonous carbon biogeochemistry in alpine and subarctic lakes. For instance, tree species composition, changes in soil microbial organic matter processing and alterations of bioclimatic conditions (e.g., precipitation intensity, frequency and timing and snow cover duration) all may shape the future lake carbon biogeochemistry. Other relevant conditions, including the effects of freeze-thawing or UV photodegradation warrant additional experimental work.

## Biodegradation and bacterial function

DOC concentration was measured with a Shimadzu TOC-L for all long-term incubation samples and the short-term incubation samples at the subarctic site and with a Shimadzu TOC-V$_{CPH}$ for the short-term incubation samples at the alpine site. DOC degradation kinetics in the long-term incubation were modeled using a reactivity continuum model, which considers each compound as a reactive type assumed to degrade following first-order kinetics. We used a Gamma distribution as the initial distribution of reactivities, according to Koehler et al.[47] and modeled the relative DOC concentration (see also Supplementary Fig. S3):

$$(DOCt/DOC0) = \exp(\alpha/\alpha + t)\,\nu \tag{1}$$

Model parameters were estimated using the nonlinear regression package *nlme* in R[48]. Oxygen was measured with a Microx system (PreSens) to determine bacterial respiration and ensure oxic conditions throughout the experiment. Experiments started under $O_2$ saturation and the minimal value reached at the end of the incubation was 6.9 mg $O_2$/L (77.2%), a minor change in redox conditions. Bacterial respiration rates were obtained as described in Supplementary Fig. S11. Bacterial carbon production was determined via incorporation of ³H-leucine according to Smith and Azam[49]. For this, samples were incubated with L-³H- leucine (Perkin-Elmer, USA, specific activity 161 Ci mmol⁻¹) for 1 h under laboratory conditions at a final leucine concentration of 40 nM. Bacterial growth efficiency was calculated as the bacterial carbon production divided by the sum of bacterial carbon production and respiration.

## DOM characterization

Changes in dissolved organic matter composition upon addition of soil extract were tracked using electrospray ionization Fourier-transformed ion cyclotron resonance mass spectrometry (FT-ICR MS) and optical measurements. FT-ICR MS measurements were performed on samples obtained from in situ experiments. Triplicate samples were filtered and acidified to pH 2 using HCl prior to solid-phase extraction (SPE) using 100 mg PPL cartridges (Bond Elut, Agilent)[50]. The extraction was performed in the field laboratories so there was no storage of the samples prior to their SPE-extraction. Thus, for FT-ICR MS, we analyzed SPE-DOM, the most widespread method of sample pre-treatment[50]. SPE-DOM represents a subset of the entire pool of DOM, which can lead to some underrepresentation of the most hydrophilic fraction[51]. Here, the use of multiple approaches to assess degradation including the whole DOM pool is used to validate the interpretation of the results obtained at the molecular level. Methanol eluates were preserved at −20 °C until analysis, when they were re-filtered through 0.2 μm PTFE syringe filters and diluted 1:1 (v/v) with ultrapure water. Analyzes were run on a 15 T ultra-high resolution FT-ICR MS (Bruker Daltonics, Bremen, Germany) in Oldenburg, Germany. Samples were injected by electrospray ionization in negative mode (−4 kV) at a rate of 3 μL min⁻¹, with a drying gas flow at 220 °C. Mass spectra were collected between 150 and 2000 *m/z* over 500 scans and internally calibrated using a list of 51 expected compounds. Peaks were identified using the Bruker Daltonics Data Analysis Software. The detection limit was calculated according to Riedel et al.[52]. Molecular formulae were considered over the ranges $C_{1-40}H_{1-82}O_{1-40}N_{0-4}S_{0-1}P_{0-1}$ under the conditions $O \leq C$; $0.3 \, C \leq H \leq 2 \, C + 2$; $O > (2 \, P + S)$. Regarding heteroatoms, only molecules with these combinations: N*S*P, $N_2$*S, $N_3$*S, $N_4$*S, $N_2$*P, N*S$_2$, $N_3$*P, N*S$_2$, $N_2$*S$_2$, $N_3$*S$_2$, $N_4$*S$_2$, S$_2$*P were considered. Unassigned masses were not further included in the data treatment. Sample SAA_S1_A_T72 was lost and replicates GKS_S1_A_T0 and GKS_S2_A_T72 were removed as outliers based on multivariate ordination (Supplementary Fig. S5). Peaks found in the blank samples, contaminant masses registered for commercial use and unassigned peaks were removed from the dataset. Peaks with an intensity <0.01% the sum of intensities were set to zero. Finally, all peaks were normalized by the sum of the intensities and only peaks present in at least two out of the three replicates were included. Nominal oxidation state of carbon (NOSC) was calculated according to Riedel et al.[53]. We classified molecular formulae into four broad chemical compound classes, based on O/C, H/C, and the modified aromaticity index (AI$_{mod}$)[54]. These classes, according to Merder et.al.[55] were "saturated" (H/C ≥ 2), "unsaturated" (H/C ≥ 1.5), "highly unsaturated" (H/C < 1.5, AI$_{mod}$ < 0.5), and "aromatics" (0.5 < AI$_{mod}$ < 0.67).

## Optical measurements and PARAFAC model

Optical measurements were performed following the guidelines in Murphy et al.[56]. Briefly, absorbance spectra (200 to 800 nm) were measured at 1 nm intervals with a Lambda 40 spectrophotometer and

UV WinLab software (Perkin-Elmer, Waltham, USA) and fluorescence excitation-emission matrices (EEM) with a spectrofluorometer (SPEX Fluoromax-4, Horiba Jobin Yvon), using a 1 cm quartz cuvette and Milli-Q water as blank. Excitation wavelengths ranged from 250 to 445 nm at intervals of 5 nm, and emission wavelengths from 300 to 600 nm at increments of 4 nm. Samples were corrected for instrument-specific biases, blank-subtracted, inner filter effect corrected[57] and normalized to Raman peak area using the FDOMcorr toolbox for MATLAB (Mathworks, Natick, MA, USA). Parallel Factor Analysis (PARAFAC) was used to identify the main components of the EEMs[58]. Six components were identified, components C2, C1, and C4 (ordered from longer to shorter emission wavelength) presented a humic-like character, related to terrestrial sources and vegetation, while components C3, C5, and C6 corresponded to tryptophan-like, amino-acid-like and tyrosine-like fluorophores (see details in Supplementary Figs. S1 and S2).

### Data treatment/statistical analysis

FT-ICR MS compound enrichment was calculated as

$$(\text{Intensity}_{\text{Treatment}} - \text{Intensity}_{\text{Control}})/\text{Intensity}_{\text{Control}} \times 100 \quad (2)$$

When $\text{Intensity}_{\text{Control}}$ was equal to zero but $\text{Intensity}_{\text{Treatment}} > 0$, enrichment was considered to be 100%. We developed a measure of chemodiversity that incorporates chemical similarity. An underlying assumption of this approach is that molecular compounds which share chemical properties have similar functional consequences. To obtain relational information based on chemical properties, we clustered the DOM molecular formulae using Gower's distance and compound-specific chemical properties (O/C, H/C, molecule mass, $AI_{\text{mod}}$ and Double Bound Equivalents). We also included the assignment of DOM compounds into categories (oxygen-poor highly unsaturated, oxygen-rich highly unsaturated, polyphenols, oxygen-poor polyphenols, saturated fatty acids (CHO), saturated fatty acids (CHOX), oxygen-rich unsaturated aliphatic and oxygen-poor unsaturated aliphatic compounds) into this cluster analysis. Next, we obtained a dendrogram depicting chemical similarity among the DOM compounds (Supplementary Fig. S9). In analogy to a phylogenetic tree that depicts the phylogenetic relatedness of taxa in a community, this dendrogram depicts classes of chemical compounds with shared chemical properties (see also ref. 31). We used a diversity measure based on path-length in this dendrogram of chemical similarity. However, since overall pathlength depends on the number of compounds, we randomly selected 3000 compounds from each sample and calculated chemodiversity. This procedure was repeated 100 times and the results of all iterations are shown as a boxplot (Fig. 2). In contrast to other measures of chemodiversity (e.g., Rao's quadratic entropy[30]) our estimates are independent of peak intensity, thus reflecting presence or absence of chemically distinct moieties in DOM pools. Yet, this novel chemodiversity index was correlated to Rao's quadratic entropy estimate ($r = 0.44$, Supplementary Fig. S10). Our measure of chemodiversity is initially dependent on the detection limit, a dependence accounted for by randomly subsampling DOM moieties (i.e., rarefying) from samples with higher total peak intensity to match samples with lower total peak intensity and then by the random selection of the 3000 molecular formulae subset.

We used linear mixed effects models to assess differences between treatments during the long-term incubations (Supplementary Table S1). In order to assess the degradation and accumulation/production of individual compounds during short-term incubations, we followed the procedure described in ref. 59 to obtain first-order decay rates. Here, positive rates indicate that the compound (as a degradation by-product) is accumulating, whereas negative rates indicate that the compound was degraded (Supplementary Figs. S6 and S7).

### Reporting summary

Further information on research design is available in the Nature Portfolio Reporting Summary linked to this article.

## Data availability

FT-ICR MS data and other data generated or analyzed during this study have been deposited at Zenodo under https://doi.org/10.5281/zenodo.10578476. The PARAFAC model for the fluorescence spectroscopy data can also be found at the OpenFluor database (https://openfluor.lablicate.com/) under accession number: 18555 where it can be directly compared with similar models.

## Code availability

Source code for the generation of the figures and associated data processing have been deposited along with the raw data in the same Zenodo repository (https://doi.org/10.5281/zenodo.10578476).

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

## Acknowledgements

This research was supported by the European Union Seventh Framework Program [FP7/2007–2013] under grant no 262693 [INTERACT] to C.R. and the Austrian Science Fund (FWF) through project P-24098-B22 to MTP and RS. L.T. acknowledges funding from the Knut and Alice Wallenberg Foundation through grant 2018.0191 and from the Swedish

Research Council through grant 2018-04524. N.C. acknowledges funding from the Wenner-Gren Foundation and is at present supported by the "Ramón y Cajal" fellowship RYC-2021-033714-I from the Ministry of Science and Innovation of Spain. We thank Antero Järvinen, Rauni Partanen, Oula Kalttopää, and Pirjo Hakala from Kilpisjärvi Biological Station for assistance with the experiment in Finland and Laurent Moya, Fabian Drewes, and Jolien Scholten for help during the experiment in Austria. We also thank Katrin Klaproth for all the help, guidance, and work when analyzing the samples at the ICBM, Oldenburg.

## Author contributions

According to CRediT (Contributor Roles Taxonomy): N.C.: Conceptualization, methodology, investigation, formal analysis, data curation, writing—original draft, writing—review & editing, visualization: C.R.: Conceptualization, methodology, investigation, funding acquisition: C.V.: formal analysis, data curation: M.T.P.: Funding acquisition: T.D.: Methodology, resources, writing—review & editing: L.T.: Resources, writing—review & editing: R.S.: Conceptualization, funding acquisition, writing—review & editing: H.P.: Conceptualization, methodology, investigation, formal analysis, data curation, writing—original draft, writing—review & editing, visualization.

## Competing interests

The authors declare no competing interests.
