## [Peer Review File · Nature Communications]

Treeline displacement may affect lake dissolved organic matter processing at high latitudes and altitudesREVIEWER COMMENTS

Reviewer #1 (Remarks to the Author):

This manuscript utilizes a range of analyses and novel data analysis approaches to assess the impact of DOM from various soils upon the microbial activity and short-term carbon processing in alpine and subarctic lakes. The soil DOM is isolated from soils at latitudes above and below the subarctic lake, and from soils at elevations above and below the alpine lake. These soils were chosen to estimate the impacts of climate-induced treeline advancement upon microbial activity and carbon processing in these lakes.

The topic is quite novel, and the data analysis approaches are valuable; however, the experimental conditions, methods, and limited sample size severely limit the extrapolation of findings to real-world conditions. The extent to which the introduction of allochthonous DOM from a soil sample simulates the impact of climate-induced treeline advance is also questionable. Moreover, these limitations are neither acknowledged nor discussed in the manuscript. Additional details are provided in the following General and Line-specific comments.

Thorough consideration and discussion of these shortcomings and their potential impact on the relevance of this study for real-world conditions is required prior to publication.

General comments

- Excellent visualization of data and novel data analysis.
- Many mis-referenced Figures and Tables in SI, and some SI Figures were not referred to in the text (i.e. Fig. S2 and S5).
- Limitations due to potential impacts of methods should be discussed (e.g. filtration, acidification and SPE).
- Should be better contextualized within the broader literature associated with permafrost thaw and climate change, and associated changes in DOM properties.
- Limited time and conditions (i.e. no freeze-thaw cycles) for incubation experiments, and the associated adaptation of microbial community structure impedes extrapolation to real-world conditions.
- Generally well written, but many minor shortcomings in spelling/grammar/clarity.

Line-specific comments

Line 28: FT-ICR MS?

Lines 32–33: ...reduced overall and compound-level DOM reactivity... is a bit difficult to read. Maybe ...reductions in overall and compound-level DOM... ?

Lines 42–45: The link between the variables governing treeline position and climate change is absent, such that “consequently” is not fitting. Maybe add another sentence and/or references connecting climate change to increasing soil temperature, regions of permafrost thaw, increased precipitation, and a changing snowpack duration at higher altitudes and latitudes? Or reorganize to include some of this information that is in the next paragraph?

Lines 54–55: ...exists yet... “Yet” is redundant and could be removed for clarity. Referring specifically to “climate-change related” treeline shifts in this way also suggests that this study is only concerned with this particular type of treeline shift, and that the study has been designed to focus only on/isolate the impact from that particular cause. Suggest rephrasing.

Lines 57–59: Suggest rephrasing this sentence to improve clarity. Also may want to be more specific than “carbon biogeochemistry”, e.g. refer to changes in the sources and nature of the carbon?

Line 64: Suggest: ...vulnerability of lakes to climate change...

Line 70: Suggest: ...in-situ and laboratory experiments.

Lines 75–76: Eighty one days is not really a long-term experiment within the context of climate change, or the adaptation and evolution of microbial communities. This may be better described as a short term experiment simulating impact on the first season after the allochthonous DOM is introduced. Freeze-thaw cycles are also an important consideration for any long-term projections given their impact on DOM and microbial communities, and the limited time each year when these northern soils and lakes are not frozen.

Line 82: Couple of spelling/grammar issues here. Suggest: ...from above the treeline. ...that the experimental manipulations would result in reduced DOM processing in lakes.

Line 89: Replace hyphen (-) with “en dash” (–) in 12–15% (and for all such ranges throughout the manuscript).

Line 94: Suggest removing comma after “both”.

Line 108: Suggest adding “previously” after the comma, and adding a reference to support the assertion that humic-like fluorescence components are related with terrestrial sources. Add reference number for model in OpenFluor to caption for Figure S1. Perhaps noteworthy, the protein/polyphenol-like component C5 is similar to what has been seen in the leachates of tree leaves and leaf litter (e.g. Cuss and Guéguen, 2015, Distinguishing dissolved organic matter at its origin: Size and optical properties of leaf-litter leachates)

Line 121: Suggest removing comma after “chemodiversity”.

Figure S3: In the caption, state/clarify the meanings of the terms in the Legend (also correct S^{Above} vs. S^{above}). This figure is erroneously referred to as Figure S2 in the text (line 136). Figure S2 also appears to have an additional line separating compound groups that is not described in Figure 1 (main text). Here and in other Figures, “-1” should be in superscript format in axis labels.

Figure S8: Turquoise is misspelled in the caption.

Line 142: Define lme.

Figure 2: Caption: Based on the figure, the S-Above treatments and the S-Below treatments converged to similar levels of chemodiversity as the Control in the subarctic lake, and neither really converged to similar levels as the control in the alpine lake, but S-Above was generally more similar in both cases. Better to say “...more similar levels...”?

Figure 3: Suggest adding lines to separate compound classes.

Line 155: Suggest removing the first “the” on this line.

Line 159: Figures 3 and S7?

Line 161: SI Table 3.2 seems to be missing.

Line 194: Suggest removing comma.

Lines 193–200: The limitations of this study with respect to being an accurate representation of the long-

term impacts of treeline advancement on DOM quality and degradation should be discussed, and the work should be compared with any other similar studies that e.g. use other methods to assess changes in carbon quality in subarctic and alpine lakes resulting from climate-induced changes.

Line 204: Define a.s.l.

Lines 223–225: Please state whether the extracts were filtered, and through what pore size(s) of membranes. This is important information for assessing the contribution of bacteria and fungi from the inocula, and also for the relative size of the DOM inoculum compared to lake water which was filtered at 0.2 μm in the laboratory incubations.

Lines 227–244: Filtering at 1.0 or 1.2 μm can remove substantial fungal species that work in concert with the bacterial population to degrade allochthonous organic matter in freshwaters (e.g. <https://aslopubs.onlinelibrary.wiley.com/doi/full/10.1002/lno.11242>). The potential impacts of removing this microbial component on the experiment and its relevance for the natural environment should be noted and discussed.

Line 252: Suggest replacing “along” with “throughout”.

Line 253: Suggest replacing “are” with “were” so that the tense is consistent.

Line 264: Humic acids are insoluble at pH 2, such that anything remaining dissolved at pH 2 will be fulvic acids. Various SPE approaches are also known to be selective/biased for certain types of compounds (e.g. Chen et al. (2016), *Anal. Bioanal. Chem.* 408: “A careful examination via FT-ICR-MS revealed that the formulas lost by the SPE might be all DOM source dependent. Nevertheless, the dominant missing compound groups were identified to be the tannins group with high O/C ratios (>0.7), lignins/carboxyl-rich alicyclic molecules (CRAM), aliphatics with high H/C >1.5, and heteroatomic formulas”). Please remark on how these may impact/limit the complete characterization of the DOM when the SPE extractant is analyzed using FT-ICR MS.

Lines 275–276: Numerical values should be in subscript format in heteroatom molecular combinations.

Line 278: The multivariate ordination appears to be missing from the SI.

Line 280: Suggest replacing “were removed, as well as” with “and”.

Lines 281–283: Incorrect spelling/grammar. Maybe separate into two sentences?

Line 284: Please elaborate on the compound categories that were used, or make reference to the location in this text where it was discussed if applicable.

Line 295: Please state the number of EEMs, as well as the methods that were used to determine the number of components in PARAFAC (e.g. split-half analysis, core consistency diagnostic, residuals analysis, etc.).

Line 298: C5 is referred to as tryptophan-like or tyrosine-like here, but shown as amino acid like in the SI. Please clarify.

Line 328: “SI Fig. 6–7” should be “Fig. 3 and SI Fig. 7”?

Reviewer #2 (Remarks to the Author):

Overall

1. This study brings together multiple lines of evidence from advanced chemical analyses (FT-ICR/MS) of samples collected from the experimental manipulations, to fluorescence spectroscopy, to kinetics modeling. These approaches reveal important new information about DOM quality in lakes affected by treeline advancement and the impacts on lake biogeochemistry. I am convinced, from some of the evidence clearly conveyed, that soils below treeline provided less biologically labile DOM to the two lakes studied, that likely reduces or negatively impacts C cycling by bacterial communities compared to soils above treeline. However, some of the evidence is not clearly conveyed, and requires re-reading and produces confusion, so the authors should carefully rewrite those sections. Nevertheless, the research, which was undoubtedly a major undertaking in terms of field work and laboratory experimental and analytical efforts, is sound and fills an important gap in our knowledge of how climate change and advancing treelines will affect our most sensitive lake systems.

2. The authors make a strong case for the research, citing the importance of the taiga-tundra zone and large number of lakes above treeline. I would add that many of these lakes are also drinking water reservoirs or supply water to downstream communities, so the importance for water quality is also worth discussing. More humic DOM has greater potential to form disinfection byproducts (for water treatment systems using chlorine for disinfection). Also, the input of soil-derived OM in high mountain lakes is likely to include organic pollutants.

It also seems that regional differences in N and P deposition may be affecting the above treeline results, judging from the diverging patterns in n-containing and p-containing compounds of the two lakes. This could be mentioned/added to the discussion.

3. To make the paper more accessible to the reader, it would be useful to provide context for the saturated, unsaturated, highly unsaturated, and aromatic categories early on in the manuscript. The first time the categories are introduced, L97-103, the information is difficult to make sense of. An introductory sentence is needed in this paragraph to tell the reader (who is not likely to be an expert in FT-ICR/MS) one: that there are several categories that differ based on the H/C ratio or aromaticity, and two: why these categories are important.

4. The use of “chemodiversity” is a novel concept. But is there some advantage of chemical diversity or some benefit to ecosystem health? Please describe. Also, from the methods, the peak intensity is not part of this calculation, and that may be beneficial – it is therefore more like a Shannon diversity measure. However, the diversity will be a function of the detection limit. I.e., compounds below the detection limit will not be counted, or if the detection limits are subject to any variability, this can greatly affect the diversity metric. So it is important that the extractions and volumes used are consistent among samples. I did not see this mentioned in the methods – please add or clarify.

5. My major issues are with some of the writing, which lacks smooth transitions, and some passages (eg., lines 133-138) seem disconnected from the flow of the manuscript. Also, some sentences seem to abruptly end before getting to why the information is relevant. It is important for the authors to further develop some of the connections and make the argument toward the conclusions more clear and compelling.

Detailed comments

L62: comma after 60N,

L83: “would result”

L102 and 103: this is confusing, which lake is this referring to?

L106-110: the support from fluorescence spectroscopy is nice to see.

L130-132: The below treeline soils do not seem to increase chemodiversity in what was discussed above, so I don't see how they introduce chemically novel compounds. This should be reworded.

L133-138: what are reactivity continuum models? These have not been introduced yet. This whole paragraph seems out of place and the results given do not link to any hypothesis. There is a need for clarifying the connection of this result to reactivity of the lakes – it is not clear.

L147-149: some additional information is needed here to help the reader understand the importance. Perhaps by better describing what “effect” was similar in L145. And in L146-147, what were the soil amendment effects? These need to be stated more clearly.

L154-156: This passage is very clear and well written. More descriptive text, used in the earlier passages, is needed in this paragraph to clarify this message.

L164:, what type of decay did they follow? Or was there no decay? Or was it linear? More is needed here.

L165: add “...this, the compounds that positively correlated...”

L168: “compounds with saturation being more labile” is not clear. Do the authors mean that there are examples of saturated compounds being more biologically labile?

L178-179: this seems to contradict findings presented in L164 about very few compounds of this type following a first order decay curve. Please explain or reword.

L191: chemically novel moieties originally seemed like they would add to the chemical diversity, but here it seems that these represent recalcitrant compounds that would not participate in C cycling in the lakes. This idea, and probably the definition of “chemically novel” needs to be provided much earlier.

L200: Can any of the consequences be elaborated upon further? The vague nature of this passage makes it difficult to see the impact of the work. Do you expect reduction in CO₂ emissions during biodegradation? More storage of C in alpine lakes? I realize the consequences is outside of the scope of the research, but to make a greater impact, it would be useful to make a link to how this specific climate change impact (treeline advancement) affects C sequestration/emissions/feedbacks/accounting. Some foodweb impacts also could be alluded to.

L246: The biodegradation kinetics seem somewhat of a blackbox that are not adequately described in the methods. The SI should show the DOC concentrations with time and the first order model that was plotted for each experiment. Or some other visualization of the first order rates that make the y-axis of fig S3. How were these rates obtained?

L253: remove extra period before “Bacterial”

L455: add space between “than” and “100%”

Figure S3: the lines are only shown. It would be important to add the symbol that corresponds to the decay coefficient on the date it was measured.

Reviewer #3 (Remarks to the Author):

The manuscript addresses environmentally-important issue and potentially fits the scope of the journal. The results are adequately interpreted and the conclusions might have sizable impact in the community of earth scientists working on lake chemical and microbial composition change in the response to climate warming.

However I have serious issues on methodology and potential biases linked to choice of studied objects and experimental methods. I cannot state at this point, whether these issues are not described in the paper for the sake of keeping the restricted word count, or they were simply overlooked during planning of the research design.

- 1) The authors chose two highly contrasted lakes. Comparison between limestone-dominated and crystalline rock-dominated catchment is not warranted. These rocks would provide totally different inorganic (and possibly, organic) C cycle in the lake water column and between the lake and its catchment. The authors should use the lakes of similar lithology
- 2) Representability of these two lakes is not justified. A comparison between them might not be valid given that the size of lakes (1.7 and 30 ha) is drastically different
- 3) Experimental methodology. Justification for only 3 days of exposure in the field is inconsistent with other numerous mesocosm-level experiments in lakes (including those performed by some senior authors of the manuscript)
- 4) Redox control during incubation experiments is not performed or not described. If the redox conditions are different from those in the natural settings (and between the field and laboratory incubations), results have little relevance to experimental modeling of climate warming effect on lakes
- 5) Soil depth used for sampling and soil type (according to WRB classification, for example) are not provided
- 6) Sampling storage prior to FT-ICR MS analyses is not described. Were the sampled frozen? What about storage artifacts between the sampling (especially in the field), transport, and the analyses

REVIEWER COMMENTS

Reviewer #1 (Remarks to the Author):

This manuscript utilizes a range of analyses and novel data analysis approaches to assess the impact of DOM from various soils upon the microbial activity and short-term carbon processing in alpine and subarctic lakes. The soil DOM is isolated from soils at latitudes above and below the subarctic lake, and from soils at elevations above and below the alpine lake. These soils were chosen to estimate the impacts of climate-induced treeline advancement upon microbial activity and carbon processing in these lakes.

The topic is quite novel, and the data analysis approaches are valuable; however, the experimental conditions, methods, and limited sample size severely limit the extrapolation of findings to real-world conditions. The extent to which the introduction of allochthonous DOM from a soil sample simulates the impact of climate-induced treeline advance is also questionable. Moreover, these limitations are neither acknowledged nor discussed in the manuscript. Additional details are provided in the following General and Line-specific comments. Thorough consideration and discussion of these shortcomings and their potential impact on the relevance of this study for real-world conditions is required prior to publication.

Thank you very much for your comments, which we think have improved our manuscript, as well as for your time and work invested in this revision. We agree with you regarding the limitations for extrapolating to real-world conditions. However, we abstain from any extrapolation and argue that this is (or should be) the case with nearly every experimental manipulation, which, in one way or another, reduce complexity in order focus on a single effect. Here, we experimentally evaluated potential changes in soil-derived organic matter composition with the predicted advancement of the treeline. However, it is clear, that many other, internal and external factors relevant for the degradation of allochthonous organic matter may change simultaneously with future climate conditions (please also see our response to some of these factors below). Given that we performed these experiments in two locations, fully replicated and measured decomposition in short- and long-term incubations, we cordially disagree about the argument on limited sample size. However, we agree that potential limitations should be discussed and we have thus, added to the revised manuscript caveats regarding confounding changes and experimental limitations. Yet, we also defend our experimental approach, which we think represents a fine balance between abstraction and realism (i.e. maintaining the natural complexity of allochthonous organic matter).

General comments

- Excellent visualization of data and novel data analysis.

Thank you very much for this comment.

- Many mis-referenced Figures and Tables in SI, and some SI Figures were not referred to in the text (i.e. Fig. S2 and S5).

Please note that in many occasions we referred to a section of the SI (e.g. Supplementary Information 2) and not to a specific figure (e.g. SI Figure S3). We realize that this was not clear and have corrected this in the new version with a new organization of the Supplementary

Information tables, figures and details on methods (see Supplementary Information track-changes file).

- Limitations due to potential impacts of methods should be discussed (e.g. filtration, acidification and SPE).

As the reviewer points out, sample treatment may affect DOM. Filtering samples excludes particles above the filter's nominal pore size and we clearly state these parameters in the methods. Regarding acidification and SPE, we understand the reviewer specifically refers to FT-ICR-MS. We have now clarified that we refer to the solid-phase extracted dissolved organic matter (SPE-DOM) for those analyses. SPE using PPL cartridges is a broadly used method (Dittmar et al. 2008; Kellerman et al. 2021), and it is widely recognized that SPE-DOM represents a sub-set of DOM, which includes the most non-polar DOM species through to highly polar molecules, but not the smallest polar molecules such as short chain organic acids and free amino acids (Hawkes et al. 2015). Thus, focusing on SPE-DOM for the molecular characterization can contribute to some homogenization across samples (Wunch et al. 2018) and to an underrepresentation of the most hydrophilic fraction (i.e. even to a potential underestimation of biodegradability; Grasset et al. 2023). That could limit the results on molecular compositional change, as degradation of some polar compounds might escape detection, but it will not impact the individual molecular models on degradation or the degradation kinetics of the complete DOM pool (i.e. based on DOC concentration). Additionally, we do not expect soil DOM to be as affected by extraction as, for instance, algal material (Catalán et al. 2020). We have extended the text to clarify these limitations in the methods (L317-322).

Dittmar, T. The molecular level determination of black carbon in marine dissolved organic matter. *Org. Geochem.* 39, 396–407 (2008).

Kellerman, A. M. et al. Molecular Signatures of Glacial Dissolved Organic Matter From Svalbard and Greenland. *Global Biogeochem. Cycles* 35, (2021).

Hawkes, J. A., Hansen, C. T., Goldhammer, T., Bach, W. & Dittmar, T. Molecular alteration of marine dissolved organic matter under experimental hydrothermal conditions. *Geochim. Cosmochim. Acta* (2015).

Wünsch, U. J. et al. Quantifying the impact of solid-phase extraction on chromophoric dissolved organic matter composition. *Mar. Chem.* (2018) doi:10.1016/J.MARCHEM.2018.08.010.

Grasset, C. Hydrophilic species are the most reactive components of freshwater dissolved organic matter. 0–20 (2023) doi:10.1021/acs.est.3c02175.

Catalán N., Pastor A. et al. 2020 The relevance of environment vs. composition on dissolved organic matter degradation in freshwaters. *Limnology and Oceanography*, doi: 10.1002/lno.11606

- Should be better contextualized within the broader literature associated with permafrost thaw and climate change, and associated changes in DOM properties.

We agree that permafrost thaw may indeed be an important covarying factor for future lake DOM processing in alpine and polar regions. Our study areas are mountainous with limited carbon deposits except in mires, and with already discontinuous permafrost. We have revised the text to better reflect those aspects as well as the link with permafrost thaw and climate change in general (L45-51 and L76).

- Limited time and conditions (i.e. no freeze-thaw cycles) for incubation experiments, and the associated adaptation of microbial community structure impedes extrapolation to real-world conditions.

These are indeed important points. We agree that time scales are important for understanding DOM processing in freshwater ecosystems. In fact, we performed both long and short-term incubations, to avoid to some extent bias due to limited experimental time (Casas-Ruiz et al. 2017; Catalán et al. 2016). However, to better acknowledge the importance of time-scales, we added a sentence on the value of high-frequency monitoring (*in situ* or remotely as recently proposed) (Battin et al. 2023). We consider freeze-thawing to be a very relevant factor which may change simultaneously with the treeline in a future climate. We thank you for pointing this out – particularly because it links previous work on permafrost thaw and DOM processing in the Arctic region (Wauthy et al. 2018) to expected climate-driven change. Note, also, that our experimental design includes soil-derived carbon from above the treeline, thus we are able to at least partially, assess that effect. We have clarified these points in the methods section (L246-250).

Battin, T. J., R. Lauerwald, E. S. Bernhardt, and others. 2023. River ecosystem metabolism and carbon biogeochemistry in a changing world. *Nature* **613**.

Casas-Ruiz, J. P., N. Catalán, L. Gómez-Gener, and others. 2017. A tale of pipes and reactors: Controls on the in-stream dynamics of dissolved organic matter in rivers. *Limnol. Oceanogr.* doi:10.1002/lno.10471

Catalán, N., R. Marcé, D. N. Kothawala, and L. J. Tranvik. 2016. Organic carbon decomposition rates controlled by water retention time across inland waters. *Nat. Geosci.* **9**. doi:10.1038/ngeo2720

Wauthy, M., M. Rautio, K. S. Christoffersen, L. Forsström, I. Laurion, H. L. Mariash, S. Peura, and W. F. Vincent. 2018. Increasing dominance of terrigenous organic matter in circumpolar freshwaters due to permafrost thaw. *Limnol. Oceanogr. Lett.* **3**: 186–198. doi:10.1002/LOL2.10063

- Generally well written, but many minor shortcomings in spelling/grammar/clarity.

We are grateful for your efforts to identify these shortcomings. We have carefully revised the manuscript accordingly.

Line-specific comments

Line 28: FT-ICR MS?

We have corrected this miss-spelling

Lines 32–33: ...reduced overall and compound-level DOM reactivity... is a bit difficult to read. We changed this to “associated with reductions in bulk and compound-level DOM reactivity” (L32).

Lines 42–45: The link between the variables governing treeline position and climate change is absent, such that “consequently” is not fitting. Maybe add another sentence and/or references connecting climate change to increasing soil temperature, regions of permafrost thaw, increased precipitation, and a changing snowpack duration at higher altitudes and latitudes? Or reorganize to include some of this information that is in the next paragraph?”

We have added a line and additional references to clarify this (L45-51).

Lines 54–55: ...exists yet... “Yet” is redundant and could be removed for clarity. Referring specifically to “climate-change related” treeline shifts in this way also suggests that this study is only concerned with this particular type of treeline shift, and that the study has been designed to focus only on/isolate the impact from that particular cause. Suggest rephrasing.

We removed “Yet”. We changed “climate-change related treeline shifts” to “treeline shifts” to incorporate other types of tree line shifts (L55-57).

Lines 57–59: Suggest rephrasing this sentence to improve clarity. Also may want to be more specific than “carbon biogeochemistry”, e.g. refer to changes in the sources and nature of the carbon?

The sentence has been re-phrased accordingly (L58-L62).

Line 64: Suggest: ...vulnerability of lakes to climate change...

Modified accordingly

Line 70: Suggest: ...in-situ and laboratory experiments.

Revised correspondingly

Lines 75–76: Eighty one days is not really a long-term experiment within the context of climate change, or the adaptation and evolution of microbial communities. This may be better described as a short term experiment simulating impact on the first season after the allochthonous DOM is introduced. Freeze-thaw cycles are also an important consideration for any long-term projections given their impact on DOM and microbial communities, and the limited time each year when these northern soils and lakes are not frozen.

This is an interesting point. We believe it is important to clearly differentiate here between the “instantaneous” measurement of DOM decomposition and long-term environmental changes induced by climate change. The 81 days of incubations refer to the measurement of DOM decomposition, and for DOM decomposition assays, 81 days of incubation is indeed rather long-term (compared to e.g. 12 days of incubations; Roehm et al. 2009). We deem these time-scales to be sufficiently long to be able to robustly quantify decomposition kinetics with low k , as in natural conditions, the half-lives of organic carbon in lakes is of 835 ± 2081 days (Catalán et al. 2016). Thus, we do believe that in terms of the impact of new terrestrial DOM compounds inputs on lake C processing the time scale is reasonable.

However, we agree, in order to address climate-change related long-term modifications of C biogeochemistry, one would like to perform other kind of experiments, such as whole-ecosystem manipulation experiments or long-term mesocosm studies covering at least a complete hydrological year. Such experiments are clearly beyond the scope of our study. For instance, in line with your comment, it could be expected that the spring freshet will be an important vector of new forest derived inputs, and accordingly, that the effect will be specifically important during summer months (the approximate duration of our experiment). We acknowledge that DOM processing during winter is not addressed with this approach, and added that in the text (L213-219).

Catalán, N., R. Marcé, D. N. Kothawala, and L. J. Tranvik. 2016. Organic carbon decomposition rates controlled by water retention time across inland waters. *Nat. Geosci.* 9. doi:10.1038/ngeo2720

Roehm, C. L., R. Giesler, and J. Karlsson (2009), Bioavailability of terrestrial organic carbon to lake bacteria: The case of a degrading subarctic permafrost mire complex, *J. Geophys. Res.*, 114, G03006, doi:10.1029/2008JG000863

Line 82: Couple of spelling/grammar issues here. Suggest: ...from above the treeline. ...that the experimental manipulations would result in reduced DOM processing in lakes.

Modified according to the suggestions.

Line 89: Replace hyphen (-) with “en dash” (–) in 12–15% (and for all such ranges throughout the manuscript).

We have replaced them.

Line 94: Suggest removing comma after “both”.

Modified accordingly.

Line 108: Suggest adding “previously” after the comma, and adding a reference to support the assertion that humic-like fluorescence components are related with terrestrial sources. Add reference number for model in OpenFluor to caption for Figure S1. Perhaps noteworthy, the protein/polyphenol-like component C5 is similar to what has been seen in the leachates of tree leaves and leaf litter (e.g. Cuss and Guéguen, 2015, Distinguishing dissolved organic matter at its origin: Size and optical properties of leaf-litter leachates)

We have modified the text accordingly and completed the section on the PARAFAC methods in the Supplementary Information (S. I. 1), adding multiple references including the one pointed out here by the reviewer.

Line 121: Suggest removing comma after “chemodiversity”.

Modified.

Figure S3: In the caption, state/clarify the meanings of the terms in the Legend (also correct S^{Above} vs. Sabove). This figure is erroneously referred to as Figure S2 in the text (line 136).

The figure and caption have been modified accordingly. Also, see answer above regarding the SI codes that have also been modified.

Figure S2 also appears to have an additional line separating compound groups that is not described in Figure 1 (main text). Here and in other Figures, “-1” should be in superscript format in axis labels.

We have modified the figures according to this comment.

Figure S8: Turquoise is misspelled in the caption.

Modified

Line 142: Define lme.

Definition added.

Figure 2: Caption: Based on the figure, the S-Above treatments and the S-Below treatments converged to similar levels of chemodiversity as the Control in the subarctic lake, and neither

really converged to similar levels as the control in the alpine lake, but S-Above was generally more similar in both cases. Better to say "...more similar levels..."?

We agree, modified accordingly.

Figure 3: Suggest adding lines to separate compound classes.

We have modified accordingly

Line 155: Suggest removing the first "the" on this line.

Modified accordingly.

Line 159: Figures 3 and S7?

See previous response regarding SI numeration, we have added the reference to Fig. 3.

Line 161: SI Table 3.2 seems to be missing.

We have modified the SI numeration.

Line 194: Suggest removing comma.

The sentence has been modified accordingly.

Lines 193–200: The limitations of this study with respect to being an accurate representation of the long-term impacts of treeline advancement on DOM quality and degradation should be discussed, and the work should be compared with any other similar studies that e.g. use other methods to assess changes in carbon quality in subarctic and alpine lakes resulting from climate-induced changes.

We have modified this paragraph according to this comment and the other reviewers.

Line 204: Define a.s.l.

We have added the full term.

Lines 223–225: Please state whether the extracts were filtered, and through what pore size(s) of membranes. This is important information for assessing the contribution of bacteria and fungi from the inocula, and also for the relative size of the DOM inoculum compared to lake water which was filtered at 0.2 μm in the laboratory incubations.

For each experiment, soil extracts were prepared according to Kalbitz et al. (2003). As described in Rofner et al. 2017 (doi: 10.1111/gcb.13545): 4 kg of soil was soaked in 4 L Milli-Q water for 18–19 h at 4 °C. Then, the water–soil mixture was passed through a cotton cloth and centrifuged at 2 500–10 000 g for 20–30 min. The supernatant was sequentially filtered through 3- μm (Polycarbonate, Millipore, Vienna, Austria), 1.2- μm (GF/C, Whatman, GE Healthcare, Vienna, Austria), ca. 0.7- μm (GF/F, Whatman, GE Healthcare), 0.5- μm (GF-5, Marchery Nagel, Duren, Germany), 0.45- μm (Cellulose ester, Millipore) and 0.2 μm (Cellulose acetate, Sartorius, Vienna, Austria) filters to obtain a bacteria-free soil extract (cross-checked by DAPI-staining and epifluorescence microscopy). For brevity, in the text we have modified the sentence as: "Bacteria-free soil extracts were prepared according to Kalbitz et al. (for details see Rofner et al.)" (L264-265).

Lines 227–244: Filtering at 1.0 or 1.2 μm can remove substantial fungal species that work in concert with the bacterial population to degrade allochthonous organic matter in freshwaters (e.g. <https://aslopubs.onlinelibrary.wiley.com/doi/full/10.1002/lno.11242>). The potential impacts of removing this microbial component on the experiment and its relevance for the natural environment should be noted and discussed.

We are aware that aquatic fungi are receiving increasing attention as potential actors in the aquatic carbon cycle. However, there is a lack of evidence for their quantitative importance, especially in pelagic water. For example, the paper cited by the reviewer addresses attached littoral fungi, and only isolates but not natural communities, and shows the degradation potential in culture, but not in the natural environment at natural substrate concentrations. We added a sentence to explicitly state that any significance of fungi would be underestimated in our experiments (L283).

Line 252: Suggest replacing “along” with “throughout”.

Modified as suggested.

Line 253: Suggest replacing “are” with “were” so that the tense is consistent.

Thank you - the tense has been modified accordingly.

Line 264: Humic acids are insoluble at pH 2, such that anything remaining dissolved at pH 2 will be fulvic acids. Various SPE approaches are also known to be selective/biased for certain types of compounds (e.g. Chen et al. (2016), Anal. Bioanal. Chem. 408: “A careful examination via FT-ICR-MS revealed that the formulas lost by the SPE might be all DOM source dependent. Nevertheless, the dominant missing compound groups were identified to be the tannins group with high O/C ratios (>0.7), lignins/carboxyl-rich alicyclic molecules (CRAM), aliphatics with high H/C >1.5 , and heteroatomic formulas”). Please remark on how these may impact/limit the complete characterization of the DOM when the SPE extractant is analyzed using FT-ICR MS. See previous response. Briefly here, characterizing SPE-DOM could limit the results on molecular composition changes, as some polar compounds might be degrading additionally, but it will not impact the individual molecular models on degradation or comparison of the degradation kinetics of the complete DOM pool. We have added some sentences to clarify this limitation in the methods (L317-323).

Lines 275–276: Numerical values should be in subscript format in heteroatom molecular combinations.

Modified accordingly.

Line 278: The multivariate ordination appears to be missing from the SI.

It was not missing, the NMDS was available in the previous version as Supplementary Figure 5. We have modified the SI for clarity.

Line 280: Suggest replacing “were removed, as well as” with “and”.

The sentence has been modified according to this suggestion.

Lines 281–283: Incorrect spelling/grammar. Maybe separate into two sentences?

The sentences have been modified for clarity.

Line 284: Please elaborate on the compound categories that were used, or make reference to the location in this text where it was discussed if applicable.

We have modified accordingly, adding the details and references in the text (LXX)

Line 295: Please state the number of EEMs, as well as the methods that were used to determine the number of components in PARAFAC (e.g. split-half analysis, core consistency diagnostic, residuals analysis, etc.).

All these details are available in the Supplementary information (Fig. S1.1 and S1.2). We would like to keep the main text as it is for the sake of brevity.

Line 298: C5 is referred to as tryptophan-like or tyrosine-like here, but shown as amino acid like in the SI. Please clarify.

We have clarified this accordingly in the main text and completed the interpretation of the components in the Supplementary Information (Fig. S1.1 and S1.2)

Line 328: “SI Fig. 6–7” should be “Fig. 3 and SI Fig. 7”?

They correspond to Supplementary Information figures S3.2 and S3.3. The SI numbers have been modified for clarity (see above).

Reviewer #2 (Remarks to the Author):

Overall

1. This study brings together multiple lines of evidence from advanced chemical analyses (FT-ICR/MS) of samples collected from the experimental manipulations, to fluorescence spectroscopy, to kinetics modeling. These approaches reveal important new information about DOM quality in lakes affected by treeline advancement and the impacts on lake biogeochemistry. I am convinced, from some of the evidence clearly conveyed, that soils below treeline provided less biologically labile DOM to the two lakes studied, that likely reduces or negatively impacts C cycling by bacterial communities compared to soils above treeline. However, some of the evidence is not clearly conveyed, and requires re-reading and produces confusion, so the authors should carefully rewrite those sections. Nevertheless, the research, which was undoubtedly a major undertaking in terms of field work and laboratory experimental and analytical efforts, is sound and fills an important gap in our knowledge of how climate change and advancing treelines will affect our most sensitive lake systems.

We are delighted to read such a positive evaluation of our work. We are grateful for your comments and address all points below.

2. The authors make an strong case for the research, citing the importance of the taiga-tundra zone and large number of lakes above treeline. I would add that many of these lakes are also drinking water reservoirs or supply water to downstream communities, so the importance for water quality is also worth discussing. More humic DOM has greater potential to form

disinfection byproducts (for water treatment systems using chlorine for disinfection). Also, the input of soil-derived OM in high mountain lakes is likely to include organic pollutants. It also seems that regional differences in N and P deposition may be affecting the above treeline results, judging from the diverging patterns in n-containing and p-containing compounds of the two lakes. This could be mentioned/added to the discussion.

We agree, this is a very important aspect that we had missed in the previous version of the manuscript. Some of these aspects have been added this to the introduction and discussion. (L45-62 and L209-230).

3. To make the paper more accessible to the reader, it would be useful to provide context for the saturated, unsaturated, highly unsaturated, and aromatic categories early on in the manuscript. The first time the categories are introduced, L97-103, the information is difficult to make sense of. An introductory sentence is needed in this paragraph to tell the reader (who is not likely to be an expert in FT-ICR/MS) one: that there are several categories that differ based on the H/C ratio or aromaticity, and two: why these categories are important.

Thank you for pointing this out to us. We fully agree and added these introductory sentences (L104-107).

4. The use of “chemodiversity” is a novel concept. But is there some advantage of chemical diversity or some benefit to ecosystem health? Please describe. Also, from the methods, the peak intensity is not part of this calculation, and that may be beneficial – it is therefore more like a Shannon diversity measure. However, the diversity will be a function of the detection limit. I.e., compounds below the detection limit will not be counted, or if the detection limits are subject to any variability, this can greatly affect the diversity metric. So it is important that the extractions and volumes used are consistent among samples. I did not see this mentioned in the methods – please add or clarify.

These are very interesting points thank you for bringing them up! First, to understand how important chemodiversity may be for ecosystem health, likely requires more work. However, recent work by other authors link DOM chemodiversity with microbial composition and function (Danczak et al. 2020, Tanentzap et al. 2019). This is not surprising given that DOM is a resource for heterotrophic microbes and at the same time, this pool includes the metabolic products of microbial activity. Following your comment, we have better highlighted this aspect in the revised manuscript (L121-122).

The methodological considerations of the reviewer are correct – we did not consider peak intensity in the calculation of chemodiversity. In analogy to phylogenetic diversity, which can be agnostic to species abundances, we were interested in the diversity of chemically novel compounds, rather than in the relative changes in concentration (i.e. peak intensities). You are right about that this measure of chemodiversity depends on detection limit. Indeed, we found that the number of molecular formulae detected increases with total peak intensity - despite identical extraction concentrations and volumes prior to FT-ICR MS. To account for this, we randomly sampled (i.e. rarefied) samples with higher total peak intensity to match samples with lower total peak intensity. Moreover, in our calculation of chemodiversity, we repeatedly selected a random subset of 3000 molecular formulae from the entire pool of formulae – this resampling procedure further reduces the impacts of differences in detection limit-.

We apologize that this was not clearly presented in the previous version of the manuscript and added these important points (L126 and L387-390).

Danczak, R.E., Chu, R.K., Fansler, S.J. *et al.* Using metacommunity ecology to understand environmental metabolomes. *Nat Commun* 11, 6369 (2020). <https://doi.org/10.1038/s41467-020-19989-y>

Tanentzap, A. J. *et al.* Chemical and microbial diversity covary in fresh water to influence ecosystem functioning. *PNAS* 116, 24689–24695 (2019). <https://doi.org/10.1073/pnas.1904896116>

5. My major issues are with some of the writing, which lacks smooth transitions, and some passages (eg., lines 133-138) seem disconnected from the flow of the manuscript. Also, some sentences seem to abruptly end before getting to why the information is relevant. It is important for the authors to further develop some of the connections and make the argument toward the conclusions more clear and compelling.

Thank you for pointing this out. We revised the manuscript in order to improve links and clarity of our arguments.

Detailed comments

L62: comma after 60N,
Added

L83: “would result”
Modified accordingly

L102 and 103: this is confusing, which lake is this referring to?
We have modified the text to clarify this

L106-110: the support from fluorescence spectroscopy is nice to see.
Thank you!

L130-132: The below treeline soils do not seem to increase chemodiversity in what was discussed above, so I don't see how they introduce chemically novel compounds. This should be reworded.

We assess two different aspects. While in the previous description we focus on which specific categories of compounds are enriched in each treatment, here we evaluate the overall effect on chemodiversity as defined in the methods (L126). Additionally, please check the total number of enriched compounds (Table 1), where the S-Below the treeline treatment added a major number of compounds than the S-Above one.

L133-138: what are reactivity continuum models? These have not been introduced yet. This whole paragraph seems out of place and the results given do not link to any hypothesis. There is a need for clarifying the connection of this result to reactivity of the lakes – it is not clear. We have added some lines to clarify this (L141-150). Those results are directly linked to the objective of assessing whether allochthonous DOM increases the overall DOM reactivity and the reactivity models are the tools used to assess the degradation kinetics.

L147-149: some additional information is needed here to help the reader understand the importance. Perhaps by better describing what “effect” was similar in L145. And in L146-147, what were the soil amendment effects? These need to be stated more clearly.

We have modified these lines following this comment.

L154-156: This passage is very clear and well written. More descriptive text, used in the earlier passages, is needed in this paragraph to clarify this message.

Thank you for the comment. We have modified this and the previous paragraph in light of this.

L164:, what type of decay did they follow? Or was there no decay? Or was it linear? More is needed here.

Either they did not change or had different kinetics, we did not assess that, both explanations are possible. We have added a statement to clarify this point (L178-179).

L165: add “...this, the compounds that positively correlated...”

Modified accordingly.

L168: “compounds with saturation being more labile” is not clear. Do the authors mean that there are examples of saturated compounds being more biologically labile?

Yes, we meant that. The text has been modified accordingly (L183).

L178-179: this seems to contradict findings presented in L164 about very few compounds of this type following a first order decay curve. Please explain or reword.

We have clarified that statement (L195). For the alpine site, the number of compounds that could be fit to an exponential decay curve was higher in the S-Below treatment.

L191: chemically novel moieties originally seemed like they would add to the chemical diversity, but here it seems that these represent recalcitrant compounds that would not participate in C cycling in the lakes. This idea, and probably the definition of “chemically novel” needs to be provided much earlier.

The chemical novelty refers to the fact that those compounds added by the soil below the treeline are more chemically different. That is not referring to its degradability. We have added this in the definition of chemodiversity to improve clarity (L126).

L200: Can any of the consequences be elaborated upon further? The vague nature of this passage makes it difficult to see the impact of the work. Do you expect reduction in CO₂ emissions during biodegradation? More storage of C in alpine lakes? I realize the consequences is outside of the scope of the research, but to make a greater impact, it would be useful to make a link to how this specific climate change impact (treeline advancement) affects C sequestration/emissions/feedbacks/accounting. Some foodweb impacts also could be alluded to.

Thank you for this comment. The final paragraph has been extensively modified according to this and also the other reviewer’s comments (L209-230).

L246: The biodegradation kinetics seem somewhat of a blackbox that are not adequately described in the methods. The SI should show the DOC concentrations with time and the first order model that was plotted for each experiment. Or some other visualization of the first order rates that make the y-axis of fig S3. How were these rates obtained?

We have modified the methods to extend the description of the reactivity continuum models used to describe the degradation kinetics. That model departs from the assumption that each constituent of DOM decays according to a first exponential. Thus, we will fit a distribution of probabilities shaped by two parameters α and ν . The initial distribution of reactivities most frequently chosen is the Gamma distribution [Boudreau and Ruddick, 1991; Arndt et al., 2013], which captures very well the organic matter biodegradation dynamics observed in inland waters [Koehler et al., 2012). Accordingly, there is no fit of a first order exponential to the bulk DOC concentrations with time. Even so, we have added a graph with the DOC concentrations and the predicted values in time (Fig. S 2.1) and modified the methods section for clarity (L298-L303).

Arndt, S., B. B. Jørgensen, D. E. LaRowe, J. J. Middelburg, R. D. Pancost, and P. A. G. Regnier. 2013. Quantifying the degradation of organic matter in marine sediments: A review and synthesis. *Earth-Science Rev.* 123: 53–86. doi:10.1016/j.earscirev.2013.02.008

Boudreau, B. P., and B. R. Ruddick. 1991. On a reactive continuum representation of organic matter diagenesis. *Am. J. Sci.* 291: 507–538. doi:10.2475/ajs.292.1.79

Koehler, B., E. Von Wachenfeldt, D. N. Kothawala, and L. J. Tranvik. 2012. Reactivity continuum of dissolved organic carbon decomposition in lake water. *J. Geophys. Res. Biogeosciences* 117: 1–14. doi:10.1029/2011JG001793

L253: remove extra period before “Bacterial”

Modified accordingly

L455: add space between “than” and “100%”

Modified accordingly

Figure S3: the lines are only shown. It would be important to add the symbol that corresponds to the decay coefficient on the date it was measured.

See response above. We hope that the approach and model is now clearer. We have modified this figure to improve that.

Reviewer #3 (Remarks to the Author):

The manuscript addresses environmentally-important issue and potentially fits the scope of the journal.

The results are adequately interpreted and the conclusions might have sizable impact in the community of earth scientists working on lake chemical and microbial composition change in the response to climate warming.

However I have serious issues on methodology and potential biases linked to choice of studied objects and experimental methods. I cannot state at this point, whether these issues are not described in the paper for the sake of keeping the restricted word count, or they were simply overlooked during planning of the research design.

We are very grateful to this Reviewer for these comments. The comments have helped us to broaden our perspective and have been very valuable to help us improve the manuscript.

Please see below our detailed response to your methodological/study design concerns.

1) The authors chose two highly contrasted lakes. Comparison between limestone-dominated and crystalline rock-dominated catchment is not warranted. These rocks would provide totally different inorganic (and possibly, organic) C cycle in the lake water column and between the lake and its catchment. The authors should use the lakes of similar lithology

We agree that the lithological set-up greatly influences the physico-chemical characteristics of a lake. Our choice of the two lakes was based on manifold considerations, primarily their proximity to the treeline and in regions where treeline shift is relevant: at high altitude and at high latitude. Other considerations were also key, such as having prior knowledge on these ecosystems as well as access and logistics. Thus, it is important to stress that we did not seek to have two replicated systems, but rather two examples of lakes affected by high altitude and high latitude treeline shifts. Besides different underlying bedrock, these lakes differ in other characteristics such as morphology (see comment below), catchment size or food web structure. We revised the manuscript carefully to better highlight that we are not attempting to replicate the effects of treeline advances, but rather to test if similar effects of treeline advancement on DOM processing can be found even in two contrasted ecosystems, at high latitude and at high altitude (L213-220; L234-235 and L246-250).

2) Representability of these two lakes is not justified. A comparison between them might not be valid given that the size of lakes (1.7 and 30 ha) is drastically different

This relates, partly to the previous comment. Our study design enabled us to demonstrate similar potential treeline shift effects in different ecosystems. Lake Gossenköllesee and lake Saanajaervi are well studied sites and both are sites with ongoing long-term ecological monitoring efforts. Both lakes are representative for numerous similar lakes in their area and, to our knowledge, our work is a first effort towards an understanding of potential consequences of an advancing treeline for carbon biogeochemistry in alpine and subarctic lakes. However, to assess potential and realized global effects of treeline shift will require substantial work and a large set of sampling sites. We hope our prime study will inspire such work. To acknowledge this, we have added a brief discussion in the revised manuscript (L213-220).

3) Experimental methodology. Justification for only 3 days of exposure in the field is inconsistent with other numerous mesocosm-level experiments in lakes (including those performed by some senior authors of the manuscript)

In fact, we present data from both, short and long-term (over 81 days) incubation experiments. Both, short and long-term incubations have their benefits and limitations. In short-term incubations, only the onset of decomposition and thus changes in readily available compounds may be detected. Long-term incubations may allow also the characterization of more slowly-decomposing compounds. However, microbial community dynamics (e.g. viral lysis, co-metabolism) may play an increasingly important role during long-term incubations. We agree that these are important considerations and we revised the manuscript to better explain our underlying reasoning (L253-255).

4) Redox control during incubation experiments is not performed or not described. If the redox conditions are different from those in the natural settings (and between the field and laboratory

incubations), results have little relevance to experimental modeling of climate warming effect on lakes

We did control for oxygen in the laboratory incubations (see information on bacterial respiration throughout the experiment), without falling into a range out of what is found in the natural settings. That make us think that other redox directions than respiration are unlikely. This information has been added in L305-307.

5) Soil depth used for sampling and soil type (according to WRB classification, for example) are not provided

The uppermost soil layer was sampled at six different points, and stones, roots and large organic debris were removed. This information is provided in detail in Rofner et al. 2016 GBC, we have added information about the sampled soil depth in the text (L264-265) but still refer to that manuscript for details.

6) Sampling storage prior to FT-ICR MS analyses is not described. Were the samples frozen? What about storage artifacts between the sampling (especially in the field), transport, and the analyses

Samples were directly extracted into solid phase extraction cartridges in the field laboratories. Therefore, there was no storage of the samples and we expect no artifacts in that regard. We have clarified this in the text (L320-321).

REVIEWERS' COMMENTS

Reviewer #3 (Remarks to the Author):

The authors adequately addressed all issues raised during review